# *Citrus limon* (Lemon) Phenomenon—A Review of the Chemistry, Pharmacological Properties, Applications in the Modern Pharmaceutical, Food, and Cosmetics Industries, and Biotechnological Studies

**DOI:** 10.3390/plants9010119

**Published:** 2020-01-17

**Authors:** Marta Klimek-Szczykutowicz, Agnieszka Szopa, Halina Ekiert

**Affiliations:** Chair and Department of Pharmaceutical Botany, Jagiellonian University, Collegium Medicum, Medyczna 9, 30-688 Kraków, Poland; marta.klimek-szczykutowicz@doctoral.uj.edu.pl (M.K.-S.); mfekiert@cyf-kr.edu.pl (H.E.)

**Keywords:** lemon, chemical composition, biological activity, cosmetic applications, phototoxicity, biotechnological studies

## Abstract

This review presents important botanical, chemical and pharmacological characteristics of *Citrus limon* (lemon)—a species with valuable pharmaceutical, cosmetic and culinary (healthy food) properties. A short description of the genus *Citrus* is followed by information on the chemical composition, metabolomic studies and biological activities of the main raw materials obtained from *C. limon* (fruit extract, juice, essential oil). The valuable biological activity of *C. limon* is determined by its high content of phenolic compounds, mainly flavonoids (e.g., diosmin, hesperidin, limocitrin) and phenolic acids (e.g., ferulic, synapic, p-hydroxybenzoic acids). The essential oil is rich in bioactive monoterpenoids such as D-limonene, β-pinene, γ-terpinene. Recently scientifically proven therapeutic activities of *C. limon* include anti-inflammatory, antimicrobial, anticancer and antiparasitic activities. The review pays particular attention, with references to published scientific research, to the use of *C. limon* in the food industry and cosmetology. It also addresses the safety of use and potential phototoxicity of the raw materials. Lastly, the review emphasizes the significance of biotechnological studies on *C. limon*.

## 1. Introduction

*Citrus limon* (L.) Burm. f. is a tree with evergreen leaves and yellow edible fruits from the family *Rutaceae*. In some languages, *C. limon* is known as lemon (English), Zitrone (German), le citron (French), limón (Spanish), and níngméng, 檸檬 (Chinese).

The main raw material of *C. limon* is the fruit, particularly the essential oil and juice obtained from it. The *C. limon* fruit stands out as having well-known nutritional properties, but it is worth remarking that its valuable biological activities are underestimated in modern phytotherapy and cosmetology [1].

*C. limon* fruit juice (lemon juice) has traditionally been used as a remedy for scurvy before the discovery of vitamin C [2]. This common use of *C. limon*, known since ancient times, has nowadays been supported by numerous scientific studies. Other uses for lemon juice, known from traditional medicine, include treatment of high blood pressure, the common cold, and irregular menstruation. Moreover, the essential oil of *C. limon* is a known remedy for coughs [3,4,5].

In Romanian traditional medicine, *C. limon* essential oil was administered on sugar for suppressing coughs [3]. Aside from being rich in vitamin C, which assists in warding off infections, the juice is traditionally used to treat scurvy, sore throats, fevers, rheumatism, high blood pressure, and chest pain [6].

In Trinidad, a mixture of lemon juice with alcohol or coconut oil has been used to treat fever, coughs in the common cold, and high blood pressure. Moreover, the juice or grated skin, mixed with molasses, has been used to remove excess water from the body, and the juice mixed with olive oil has been administered for womb infection and kidney stones [4]. According to Indian traditional medicine, *C. limon* juice can induce menstruation; the recommended dose for this is two teaspoons consumed twice a day [5].

Currently, valuable scientific publications focus on the ever wider pharmacological actions of *C. limon* fruit extract, juice and essential oil. They include studies of, for example, antibacterial, antifungal, anti-inflammatory, anticancer, hepatoregenerating and cardioprotective activities [7,8,9,10,11].

The pharmacological potential of *C. limon* is determined by its rich chemical composition. The most important group of secondary metabolites in the fruit includes flavonoids and also other compounds, such as phenolic acids, coumarins, carboxylic acids, aminoacids and vitamins. The main compounds of essential oil are monoterpenoids, especially D-limonene. These valuable chemical components are the reason for the important position of *C. limon* in the food and cosmetics industries [12,13,14].

The aim of this overview is a systematic review of scientific works and in-depth analyses of the latest investigations and promotions related to *C. limon* as a valuable plant species, important in pharmacy, cosmetology and the food industry. Additionally, relevant biotechnological investigations are presented.

## 2. The Genus *Citrus*

The genus *Citrus* is one of the most important taxonomic subunits of the family *Rutaceae*. Fruits produced by the species belonging to this genus are called ‘citrus’ in colloquial language, or citrus fruits. Citrus fruits are commonly known for their valuable nutritional, pharmaceutical and cosmetic properties. The genus *Citrus* includes evergreen plants, shrubs or trees (from 3 to 15 m tall). Their leaves are leathery, ovoid or elliptical in shape. Some of them have spikes. The flowers grow individually in leaf axils. Each flower has five petals, white or reddish. The fruit is a hesperidium berry. The species belonging to the genus *Citrus* occurs naturally in areas with a warm and mild climate, mainly in the Mediterranean region. They are usually sensitive to frost [2].

One of the best known and most used species of the genus *Citrus* is the lemon—*Citrus limon* (L.) Burm. f. (Latin synonyms: *C. × limonia*, *C. limonum*). Other important species included in this taxonomic unit are: *Citrus aurantium* ssp. *aurantium*—bitter orange, *Citrus sinensis*—Chinese orange, *Citrus reticulata*—mandarin, *Citrus paradise*—grapefruit, *Citrus bergamia*—bergamot orange, *Citrus medica*—citron, and many others. A team of scientists from the University of California (Oakland, California, USA) [15] analyzed the origin of several species of the genus *Citrus*, including *C. limon*. They found that *C. limon* was a plant that had formed as a result of the combination of two species—*C. aurantium* and *C. medica*. In the studies of scientists from Southwest University of China (Chongqing, China), the metabolite profiles of *C. limon, C. aurantium* and *C. medica* were evaluated using gas chromatography–mass spectrometry (GC-MS) and the partial least squares discriminant analysis (PLS-DA) score plot [16]. They proved that *C. limon* has a smaller distance between *C. aurantium* and *C. medica* in comparison with other *Citrus* species. These studies demonstrated that *C. limon* was likely a hybrid of *C. medica* and *C. aurantium,* as previously suspected [16].

Botanical classification of the species of the genus *Citrus* is very difficult due to the frequent formation of hybrids and the introduction of numerous cultivars through cross-pollination. Hybrids are produced to obtain fruit with valuable organoleptic and industrial properties, including seedless fruit, high juiciness, and the required taste. For older varieties, hybrids and cultivars, the latest molecular techniques are often needed to identify them. *C. limon*, like many other prolific citrus species, gives rise to numerous varieties, cultivars and hybrids, which are presented in Table 1 and Table 2 acc. to [17].

One of the oldest preserved botanical sources describing species of the genus *Citrus* is the “Monograph on the Oranges of Wên-chou” (in Chinese: 記 嘉 桔 錄, “Citrus records of Ji Jia”) by Han Yanzhi from 1178 [18,19]. Other historical works describing the species bearing citrus fruits are “Nürnbergische Hesperides” from 1708 and “Traité du Citrus” from 1811. Historically, one of the best known classifications of citrus species is “Histoire Naturelle des Orangers” from 1818. The American botanist Walter Tennyson Swingle (1871–1952) had a particularly significant impact on the present-day taxonomy of the genus *Citrus*. He is the author of as many as 95 botanical names of species of the genus *Citrus*. Currently, the systematics of the species of the genus *Citrus* are based on studies of molecular markers and other DNA analysis technologies still provide new information [20].

## 3. Botanical Characteristics and Occurrence of *C. limon*

*Citrus limon* (L.) Burm. f. (lemon) is a tree reaching 2.5–3 m in height. It has evergreen lanceolate leaves. Bisexual flowers are white with a purple tinge at the edges of the petals. They are gathered in small clusters or occur individually, growing in leaf axils. The fruit is an elongated, oval, pointed green berry that turns yellow during ripening. Inside, the berry is filled with a juicy pulp divided into segments (like an orange). The *C. limon* pericarp is made of a thin, wax-covered exocarp, under which there is the outer part of the mesocarp, also known as flavedo. This part contains oil vesicles and carotenoid dyes. The inner part of the mesocarp, also known as the albedo, is made of a spongy, white parenchyma tissue. The endocarp, or ‘fruit flesh’, is divided into segments by the spongy, white tissue of the mesocarp [2].

The *C. limon* tree prefers sunny places. It grows on loamy, well-drained, moist soils with a wide pH range [1,2].

The location of the original natural habitat of *C. limon* is not accurately known [1,21]. However, *C. limon* is considered to be native to North-Western or North-Eastern India [2,17].

*C. limon* is mainly recognized as a cultivated species. It has been cultivated in southern Italy since the 3rd century AD, and in Iraq and Egypt since 700 AD. The Arabs introduced *C. limon* into Spain, where it has been cultivated since 1150. Marco Polo’s expeditions also brought *C. limon* to China in 1297. It was also one of the first new species that Christopher Columbus brought in the form of seeds to the North American continent in 1493. In the 19th century, worldwide commercial production of *C. limon* began in Florida and in California. Nowadays, the USA is the largest producer of *C. limon*. Italy, Spain, Argentina and Brazil also play a significant role [17].

## 4. *C. limon* Pharmacopoeial Monographs and Safety of Use

By cold-pressing the fresh outer parts of the *C. limon* pericarp (Lat. *exocarpium*), an essential oil is obtained—the lemon oil (lat. *Citrus limon aetheroleum*, *Limonis aetheroleum*, *Oleum Citri*). The oil is colourless or yellow, and has a characteristic, strong lemon scent [21]. It is considered a pharmacopoeial raw material. Its monographs, entitled ‘*Limonis aetheroleum*’, are present in the European Pharmacopoeia 9th [22], American Pharmacopoeia [23], and in the Ayurvedic Pharmacopoeia of India [24].

Another pharmacopoeial raw material obtained from *C. limon* is the outer part of the *mesocarp*—the *flavedo*. A monograph entitled ‘*Citrus limon flavedo*’ can be found in older editions of the French Pharmacopoeia, for example, in its 10th edition from 1998 [25].

The fresh fruit of *C. limon* is officially listed for use in phytotherapy and in homeopathy in Germany. According to the German Commission D Monographs for homeopathic medicines, *C. limon* fresh fruits can be used for treating gingival bleeding and debilitating diseases [26].

*C. limon* also has a positive recommendation in the European Commission’s Cosmetics Ingredients Database (CosIng Database) as a valuable plant for cosmetics’ production [27].

The European Food Safety Authority (EFSA) classified the pericarp, fruit, and leaves of *C. limon* as raw materials of plant origin, in which there is presence of naturally occurring ingredients that may pose a threat to human health when used in the production of food and dietary supplements. EFSA has remarked that the toxic substances in these raw materials are photosensitizing compounds belonging to the furanocoumarin group, including bergapten and oxypeucedanin (Figure 1) [28].

In the American Food and Drug Administration (FDA) list, *C. limon* essential oil and extracts are classified as safe products [29].

## 5. Chemical Composition of *C. limon*

The chemical composition of *C. limon* fruit is well known. It has not only been determined for the whole fruit but also separately for the pericarp, juice, pomace, and essential oil. The compositions of the leaves and the fatty oil extracted from *C. limon* seeds are also known. Due to the large number of *C. limon* varieties, cultivars and hybrids, various research centres undertake the task of analyzing the chemical composition of the raw materials obtained from them.

The most important group of bioactive compounds in both *C. limon* fruit and its juice, determining their biological activity, are flavonoids such as: flavonones—eriodictyol, hesperidin, hesperetin, naringin; flavones—apigenin, diosmin; flavonols—quercetin; and their derivatives (Figure 2). In the whole fruit, other flavonoids are additionally detected: flavonols—limocitrin (Figure 2) and spinacetin, and flavones—orientin and vitexin (Table 3 and Table 4). Some flavonoids, such as neohesperidin, naringin and hesperidin (Figure 2), are characteristic for *C. limon* fruit. In comparison to another *Citrus* species, *C. limon* has the highest content of eriocitrin (Figure 2) [30].

Phenolic acids are another important group of compounds found both in the juice and fruit. There are mainly two such compounds in the juice—ferulic acid and synapic acid, and their derivatives. In contrast, the presence of p-hydroxybenzoic acid has been confirmed in the fruit. In the fruit, there are also coumarin compounds, carboxylic acids, carbohydrates, as well as amino acids, a complex of B vitamins, and, importantly, vitamin C (ascorbic acid) (Table 3 and Table 4) [1,12,13,31,32,33,34,35,36].

Another interesting group of compounds that are found in *C. limon* fruits are limonoids. Limonoids are highly oxidized secondary metabolites with polycyclic triterpenoid backbones. They mainly occur in citrus fruits, including lemons, in which they are found mainly in the seeds, pulp, and peel. There are predominantly two such compounds in *C. limon* fruits—limonin and nomilin (Figure 3) [37]. Studies have shown that the concentrations of the compounds of this group are dependent on fruit growth and maturation stages. Young citrus fruits contain the highest amounts of these compounds, compared to ripe ones [38].

Analysis of macroelements in *C. limon* fruit showed the presence in pulp and peel of: calcium (Ca), magnesium (Mg), phosphorus (P), potassium (K) and sodium (Na) [36].

In *C. limon* seed oil, the main ingredients are fatty acids, such as arachidonic acid, behenic acid and linoleic acid, and also tocopherols and carotenoids (Table 5) [33,35]. The latest studies showed that *C. limon* fruit pulp oil contains more fatty acids compared to other *Citrus* species, such as *C. aurantium*, *C. reticulata* and *C. sinensis.* The following fatty acids have been identified in *C. limon* pulp oil: behenic acid, erucic acid, gondoic acid, lauric acid, linoleic acid, α-linolenic acid, margaric acid, palmitic acid, palmitoleic acid, pentadecanoic acid, and stearic acid [39].

The main components of the *C. limon* essential oil are monoterpenoids. Among them, quantitatively dominant in the essential oil obtained from pericarp are: limonene (69.9%), β-pinene (11.2%), γ-terpinene (8.21%), (Figure 4), sabinene (3.9%), myrcene (3.1%), geranial (E-citral, 2.9%), neral (Z-citral, 1.5%), linalool (1.41%). In addition to terpenoids, the essential oil also contains linear furanocoumarins (psoralens) and polymethoxylated flavones (Table 6) [14,40,41].

The essential oil of the *C. limon* leaf differs in composition from oil obtained from pericarp. Its main compounds include: limonene (31.5%), sabinene (15.9%), citronellal (11.6%), linalool (4.6%), neral (4.5%), geranial (4.5%), (E)-β-ocimene (3.9%), myrcene (2.9%), citronellol (2.3%), β-caryophyllene (1.7%), terpne-4-ol (1.4%), geraniol (1.3%) and α-pinene (1.2%) (Table 6) [14,16,40,41,42,43].

## 6. Metabolomic Profile Studies

The team of Mucci et al. [35] investigated the metabolic profile of different parts of *C. limon* fruit. Flavedo, albedo, pulp, oil glands, and the seeds of lemon fruit and citron were studied through high resolution magic angle spinning nuclear magnetic resonance (HR-MAS NMR) spectroscopy. The analyses were made directly on intact tissues without any physicochemical manipulation. In *C. limon* flavedo were detected: terpenoids (limonene, β-pinene and γ-terpinene), aminoacids (asparagine, arginine, glutamine, proline), organic acids (malic acid and quinic acid), osmolites (stachydrine), and fatty acid chains and sugars (glucose, fructose, β-fructofuranose, myoinositol, scylloinositol and sucrose) (Table 3). The albedo of *C. limon* fruit showed the presence of low signals from: aminoacids (alanine, threonine, valine, glutamine), sugars (glucose, sucrose, β-fructofuranose, myoinosytol, scylloinositol and β-fructopyranose), and osmolites (stachydrine, β-hydroxybutyrate, ethanol) (Table 3). In albedo, clear signals from flavonoids were detected, such as hesperidin and rutoside, that have been identified also by high performance liquid chromatography (HPLC) analyses. Oil glands’ HR-MAS NMR composition analysis showed the presence of terpenoids (limonene, γ-terpinene, β-pinene, α-pinene, geranial, neral, citronellal, myrcene, sabinene, α-thujene, nerol and geraniol esters) and sugars (glucose, sucrose, β-fructofuranose and β-fructopyranose). The analysis of *C. limon* pulp showed the presence of aminoacids (asparagine, proline, alanine, γ-aminobutyric acid (GABA), glutamine, threonine and valine), organic acids (citric acid and malic acid), sugars (glucose, sucrose, β-fructofuranose, β-fructopyranose, myoinosytol and scylloinosytol) and osmolites (stachydrine, ethanol and methanol) (Table 3). HR-MAS NMR seeds analysis indicated that their composition is dominated by triglyceride signals (linoleic acid, linolenic acid and their derivatives), sugars (glucose and sucrose), osmolites (stachydrine) and trigonelline [35].

In another metabolomic study, the peel extracts of ripened *C. limon* fruit was characterized as containing nonfluorescent chlorophyll catabolites (NCCs) and dioxobilane-type nonfluorescent chlorophyll catabolite (DNCC) [44]. In the peels of *C. limon* fruit, four chlorophyll catabolites were detected: Cl-NCC1, Cl-NCC2, Cl-NCC3 and Cl-NCC4 [44].

The metabolomic profile of *C. limon* leaf was investigated by Asai et al. [45]. The studies showed that *C. limon* leaves contain 26 different organic acids and their derivatives (aconitic acid, 2-aminobutyric acid, 4-aminobutyric acid, ascorbic acid, benzoic acid, citramalic acid, citric acid, p-coumaric acid, ferulic acid, fumaric acid, glucaric acid, glycolic acid, 3-hydroxybutyric acid, 2-isopropylmalic acid, malic acid, malonic acid, 3-methylglutaric acid, oxamic acid, D-3-phenyllacetic acid, pipecolic acid, pyruvic acid, quinic acid, shikimic acid, succinic acid, threonic acid, urocanic acid), 21 aminoacids (alanine, γ-aminobutyric acid, anthranilic acid, asparagine, aspartic acid, glutamic acid, glutamine, glycine, histidine, isoleucine, leucine, lysine, methionine, phenylalanine, proline, pyroglutamic acid, serine, threonine, tryptophan, tyrosine, valine), and 13 sugars and sugar alcohols (arabinose, fructose, galactose, glucose, glycerol, inositol, lyxose, maltose, rhamnose, ribose, sorbose, sucrose, xylitol). Additionally, studied leaves have been exposed to stress conditions (leaves were placed in such a way that the edge of the petiole was in contact with the bottom of a glass bottle, soaked with 0.2 mM jasmonic acid and salicylic acid aqueous solutions, and incubated at 25 °C for 24 h). The content of aminoacids, such as, tyrosine, tryptophan, phenylalanine, valine, leucine, isoleucine, lysine, methionine, threonine, histidine, and γ-aminobutyric acid, was increased after this stress treatment [45].

According to Mehl et al. [46], the identification of volatile and non-volatile metabolites in *C. limon* essential oil is dependent on geographic origin and the analytical methods used. To evaluate the potential of volatile and non-volatile fractions for classification purposes, volatile compounds of cold-pressed lemon oils were analyzed, using modern methods like gas chromatography-flame ionization detector-mass spectrometer (GC-FID/MS) and fourier transform mid-infrared spectroscopy (FT-MIR), while the non-volatile residues were studied using FT-MIR with proton nuclear magnetic resonance (^1^H-NMR) and ultra-high performance liquid chromatography-quadrupole time-of-flight mass spectrometry (UHPLC-TOF-MS). The studies lead to very good differentiation and classification of samples regarding their geographic origin and extraction process modalities. The essential oil from the Italian-originated *C. limon* fruit was enriched in α-thujene, α-pinene, α-terpinene, sesquiterpenoids (i.e., β-caryophyllene) and furocoumarins (i.e., bergamottin). The essential oil from Spanish and Argentinian *C. limon* fruit was characterized by significant terpene contents, such as limonene, but differed in imperatorin, and byakangelicol contents. The studies showed that essential oil from Spanish *C. limon* fruit contained more camphor and 4-terpineol, while Argentinian *C. limon* fruit contained more sabinene and cis-sabinene hydrate [46].

The studies performed by Jing et al. [16] focused on the identification of components in the essential oil of different *Citrus* species, including *C. limon*. In general, most of the studied essential oil components were identified as monoterpenoids. The major monoterpenes in *C. limon* essential oil were: limonene (70.37%), p-mentha-3,8-diene (18.00%), myrcene (4.40%), α-pinene (3.24%), α-thujene (1.05%) and terpinolene (0.90%) (Table 6). Other monoterpenoids, which were identified as characteristic of *C. limon,* were: sabinene (0.28%), α-terpinene (0.22%), trans-muurola-4(14), 5-diene (0.18%), eucalyptol (0.12%), octanol acetate (0.03%), β-curcumene (0.03%), zonarene (0.03%), 7-epi-sesquithujene (0.02%), citronellyl acetate (0.02%), α-farnesene (0.01%) (Table 6). The shown metabolite-based profiling model can be used to clearly discriminate the basic *Citrus* species. Limonene, α-pinene, sabinene and α-terpinene were the major characteristic components of the analyzed metabolomes of *Citrus* genotypes that contributed to their taxonomy [16].

Studies performed by Masson et al. [43] deal with furanocoumarin’s and coumarin’s metabolomic profile in essential oil from *C. limon* fruit peel. *C. limon* essential oil contained large amounts of both furanocoumarins and coumarins compared to another tested *Citrus* essential oils. In *C. limon* essential oil, 13 furanocoumarins were detected (bergamottin, bergapten, byakangelicol, byakangelicin, epoxybergamottin, 8-geranyloxypsoralen, heraclenin, imperatorin, isoimperatorin, isopimpinellin, oxypeucedanin, oxypeucedanin hydrate, phellopterin) and two coumarins (citropten and herniarin) (Table 6) [43].

## 7. Biological Activity of *C. limon* Raw Materials

### 7.1. Anticancer Activity

*C. limon* nanovesicles have been isolated from the fruit juice using the ultracentrifugation method and purification on a 30% sucrose gradient, using an in vitro approach. The study showed that isolated nanovesicles (20 µg/mL) inhibited cancer cell proliferation in different tumour cell lines, by activating a TRAIL-mediated apoptotic cell death. Furthermore, *C. limon* nanovesicles suppress chronic myeloid leukemia (CML) tumour growth in vivo by specifically reaching the tumour site and by activating TRAIL-mediated apoptotic cell processes (Table 7) [47].

Another study has shown that an 80:20 methanol:water extract from lemon seeds induces apoptosis in human breast adenocarcinoma (MCF-7) cells, leading to the inhibition of proliferation. This extract showed the highest (29.1%) inhibition of MCF-7 cells in an MTT assay (Cell Proliferation Kit), compared to ethyl acetate, acetone and methanol extracts. The results suggest that aglycones and glycosides of the limonoids and flavonoids present in the 80:20 methanol:water extract may potentially serve as a chemopreventive agent for breast cancer (Table 7) [9].

### 7.2. Antioxidant Activity

It has been shown that the antioxidant activity of the flavonoids from *C. limon*—hesperidin and hesperetin—was not only limited to their radical scavenging activity but also augmented the antioxidant cellular defences via the ERK/Nrf2 signalling pathway (Table 7) [8].

In addition, vitamin C prevents the formation of free radicals and protects DNA from mutations. Studies have also shown a reduction in lipid peroxidation in seizures and status epilepticus was induced by pilocarpine in adult rats [48].

### 7.3. Anti-Inflammatory Activity

Various in vitro and in vivo studies have been conducted to evaluate hesperidin metabolites, or their synthetic derivatives, at their effectiveness in reducing inflammatory targets including NF-κB, iNOS, and COX-2, and the markers of chronic inflammation (Table 7) [8].

The essential oil from *C. limon* (30 or 10 mg/kg *p.o*.) exhibited anti-inflammatory effects in mice under formalin test by reducing cell migration, cytokine production and protein extravasation induced by carrageenan. These effects were also obtained with similar amounts of pure D-limonene. The anti-inflammatory effect of *C. limon* essential oil is probably due to the high concentration of D-limonene (Table 8) [49].

Studies by Mahmoud et al. [50] have shown the protective effects of limonin on experimentally induced hepatic ischemia reperfusion (I/R) injury in rats. The mechanism of these hepatoprotective effects was related to the antioxidant and anti-inflammatory potential of limonin mediated by the down-regulation of the TLR-signaling pathway [50].

In studies with the essential oil administered at a dose of 10 mg/kg *p.o.*, D-limonene induced a significant reduction in intestinal inflammatory scores, comparable to that induced by ibuprofen. The studies documented that D-limonene-fed rats had significantly lowered serum concentrations of TNF-α compared to untreated TNBS-colitis rats. The anti-inflammatory effect of D-limonene also involved the inhibition of TNFα-induced NF-κB translocation in fibroblast cultures. The application of D-limonene in colonic HT-29/B6 cell monolayers increased epithelial resistance. The study found evidence that IL-6 markedly decreased during dietary supplementation with D-limonene [51]. Another study showed that the oil moderately inhibited soybean 5-lipoxygenase (5-LOX) with an IC_50_ value of 32.05 μg/mL (Table 8) [52].

### 7.4. Antimicrobial Activity

Acetone extracts from *C. limon* fruits have shown inhibitory activity against the Gram-positive bacteria *Enterococcus faecalis* (MIC 0.01 mg/mL) and *Bacillus subtilis* (MIC 0.01 mg/mL), and the Gram-negative *Salmonella typhimurium* (MIC 0.01 mg/mL) and *Shigella sonnei* (MIC 0.01 mg/mL) (Table 7) [7].

Moreover, under another study, *C. limon* essential oil showed antibacterial activity against Gram-positive bacteria (*Bacillus subtilis* (MIC 2 mg/mL), *Staphylococcus capitis* (MIC 4 mg/mL), *Micrococcus luteus* (MIC 4 mg/mL)), and Gram-negative (*Pseudomonas fluorescens* (MIC 4 mg/mL), *Escherichia coli* (100% inhibition)) (Table 8) [52,53].

The *C. limon* essential oil exhibits inhibitory activity against *Staphylococcus mutans* (MIC 4.5 mg/mL) and effectively reduced the adherence of *S. mutans* on a glass surface, with adherence inhibition rates (AIR) from 98.3% to 100%, and on a saliva-coated enamel surface, for which the AIRs were from 54.8% to 79.2%. It effectively reduced the activity of glucosyltransferase (Gtf) and the transcription of Gtf in a dose-dependent manner (Table 8) [54].

Ethanol and acetone extracts from fruits of *C. limon* were active against *Candida glabrata* (MIC 0.02 mg/mL) (Table 7) [7]. On the other hand, *C. limon* essential oil ingredients, such as D-limonene, β-pinene and citral, have shown inhibitory activity against *Aspergillus niger* (MIC 90 µL/mL at 70 °C), *Saccharomyces cerevisiae* (MIC 4 mg/mL) and *Candida parapsilosis* (MIC 8 mg/mL) (Table 8) [52,55]. Another study confirmed that *C. limon* essential oil promoted a 100% reduction in the growth of *C. albicans* [56].

Moreover, other studies have shown that *C. limon* essential oil at a concentration of 0.05% inhibits *Herpes simplex* replication to the extent of 33.3% (Table 8) [57].

### 7.5. Antiparasitic Effect

The effect of *C. limon* essential oil on *Sarcoptes scabiei* var. *cuniculi* has been evaluated in vitro and in vivo. The infected parts of rabbits were treated topically once a week for four successive weeks. In vitro application results showed that *C. limon* essential oil (10% and 20%, diluted in water) caused mortality in 100% of mites after 24 h post-application. In vivo application of 20% lemon oil on naturally infected rabbits showed complete recovery from clinical signs and absence of mites in microscopic examination from the second week of treatment (Table 8) [58].

### 7.6. Anti-Allergic Effect

Aqueous extracts from the peel of *C. limon* fruits have been used to investigate their effects on the release of histamine from rat peritoneal exudate cells (PECs). The extracts inhibited the release of histamine from rat PECs induced by the calcium ionophore A23187. Heating the extracts at 100 °C for 10 min. enhanced the inhibition of histamine release. Histamine release was inhibited to the extent of 80%. The extracts potentially suppressed inflammation in mice cavity, like indometacin, a well-known anti-inflammatory drug (Table 7) [59].

### 7.7. Hepatoregenerating Effect

An ethanolic extract of *C. limon* fruits has been evaluated for its effects on experimental liver damage induced by carbon tetrachloride (CCl_4_), and the ethyl acetate soluble fraction of the extract has been evaluated for its effect on the HepG2 cell line (human liver cancer cell line). The ethanolic extract (150 mg/mL) normalized the levels of aspartate aminotransferase (ASAT), alanine aminotransferase (ALAT), alkaline phosphatase (ALP), and total direct bilirubin, which had been altered due to CCl_4_ intoxication in rats. After treatment with the extract, the level of malondialdehyde in the liver tissue was significantly reduced, hence the lipid peroxidation, and raised the level of the antioxidant enzymes superoxide dismutase and catalase. It improved the reduced glutathione levels in the treated rats in comparison with CCl_4_-intoxicated rats. The effect seen was dose dependent, and the effect of the highest dose was almost equal to the standard—silymarin. In an investigation carried out on a human liver-derived HepG2 cell line, a significant reduction in cell viability was observed in cells exposed to CCl_4_ (Table 7) [10].

Studies with *C. limon* essential oil have also shown the stimulation of liver detoxification by the activation of cytochrome P_450_ and liver enzymes (glutathione S-transferase) in chronic liver poisoning (Table 8) [21].

### 7.8. Antidiabetic Effect

Ethanol extracts from *C. limon* peel were administered orally at a dose of 400 mg/kg daily for 12 days to diabetic rats in which diabetes had been induced by the use of streptozotocin. The study showed a reduction in blood glucose, a reduction in wound healing time, and an increase in tissue growth rate, collagen synthesis, and protein and hydroxyproline levels (Table 7) [60].

Another study evaluated the antidiabetic effect of D-limonene in streptozotocin-induced diabetic rats. D-limonene was administered orally at doses of 50, 100 and 200 mg/kg body weight, and glibenclamide at a dose of 600 µg/kg body weight, daily for 45 days. The administration of D-limonene for 45 days gradually decreased the blood glucose level, and the maximum effect was observed at a dose of 100 mg/kg body weight. The activities of gluconeogenic enzymes, such as glucose 6-phosphatase and fructose 1,6-bisphosphatase, were increased, and the activity of the glycolytic enzyme, glucokinase, was decreased, along with liver glycogen, in the diabetic rats. The effect of D-limonene was more pronounced at the dose of 100 mg/kg body weight than at the two smaller doses. The antidiabetic effect of D-limonene was comparable with that of glibenclamide (Table 8) [61].

### 7.9. Anti-Obesity Activity

In a study, lemon juice was used in a low-calorie diet (‘lemon detox diet’). The diet consisted of 2 L of lemon detox juice containing 140 g ‘Neera’ syrup, 140 g lemon juice, and 2 L water per day. The study showed that *C. limon* juice caused a reduction in serum high-sensitive C-reactive protein (hs-CRP) in comparison with the *placebo* and normal diet group. Haemoglobin and haematocrit levels remained stable in the group on the lemon detox diet, while they decreased in the *placebo* and normal diet groups (Table 7) [62].

Studies have shown that D-limonene is beneficial to people with dyslipidaemia and hyperglycaemia. D-limonene at a dose of 400 mg/kg per day for 30 days promotes in male rats a decrease in LDL-cholesterol, prevents the accumulation of lipids, and affects the blood sugar level. Its antioxidant action enhances these effects. Dietary supplementation with D-limonene would restore pathological alteration of the liver and pancreas. It could help in the prevention of obesity (Table 8) [21].

### 7.10. Effects on the Digestive System

Studies have shown that D-limonene increases gastric motility and causes a reduction in nausea, neutralization of stomach acids, and relief of gastric reflux (Table 8) [21].

### 7.11. Effects on the Cardiovascular System

A study has indicated that daily intake of *C. limon* juice has a beneficial effect on blood pressure. The study was conducted on 100 middle-aged women in an island area nearby Hiroshima. Instances of lemon juice ingestion and the number of steps walked had been recorded for five months. The results indicated that daily lemon juice intake and walking were effective in reducing high blood pressure because both showed significant negative correlations with systolic blood pressure (Table 7) [63].

In vitro and in vivo studies have confirmed that *C. limon* juice (0.4 mL/kg) has a significant impact on blood pressure and on coagulation and anticoagulation factors in rabbits. In vitro tests revealed a highly significant increase in thrombin time and activated partial thromboplastin time by *C. limon*, whereas fibrinogen concentration was significantly reduced in comparison with the control; prothrombin time, however, was not affected significantly. Significant changes were observed in haematological parameters, such as amounts of erythrocytes and haemoglobin and mean corpuscular haemoglobin concentrations, in in vivo testing of *C. limon*. Bleeding time and thrombin time were significantly prolonged, and there was an increase in protein C and thrombin–antithrombin complex levels (Table 7) [11].

### 7.12. Influence on the Nervous System

The influence of *C. limon* juice on the memory of mice has been investigated using Harvard Panlab Passive Avoidance response apparatus, controlled through the LE2708 Programmer. Passive Avoidance is a fear-motivated test used to assess the short- or long-term memory of small animals, which measures the latency in entering a black compartment. Animals that were fed *C. limon* juice (0.2, 0.4 and 0.6 mL/kg) showed, in comparison with the control, a highly significant or a significant increase in latency before entering a black compartment after 3 and 24 h, respectively (Table 7) [64].

Studies have also shown that the main compound of *C. limon* essential oil—D-limonene—in concentrations of 0.5% and 1.0%, administered to mice by inhalation, has a significant calming and anxiolytic effect by activating serotonin and dopamine receptors. In addition, D-limonene has an inhibitory effect on pain receptors, similar to that of indomethacin and hyoscine (Table 8) [65].

### 7.13. Influence on Skeletal System

Studies have shown the potential use of nomilin for the inhibition of osteoclastogenesis in vitro. Cell viability of the mouse RAW264.7 macrophage cell line and mouse primary bone-marrow-derived macrophages (BMMs) with the Cell Counting Kit (Dojindo Laboratories, Kumamoto, Japan) was measured. Nomilin caused significantly decreased TRAP-positive multinucleated cell numbers (a measure of osteoclast cell numbers) when compared with the control. Moreover, the non-toxic concentrations of the compound decreased bone resorption activity and down regulated osteoclast-specific genes (NFATc1 and TRAP mRNA levels), coupled with suppression of the MAPK signaling pathway. Studies have shown the therapeutic potential of nomilin for the prevention of bone metabolic diseases such as osteoporosis [66].

### 7.14. C. limon as Corrigent in Pharmacy

In addition to the very important uses mentioned above, the oil is used in pharmacy and cosmetic formulations as a flavour and aroma corrigent, as well as a natural preservative, due to its confirmed antibacterial and fungistatic effects [21].

## 8. *C. limon* in the Food Industry

Due to the rich chemical composition of *C. limon* fruit and other lemon-derived raw materials, they have applications in the food industry and in food processing. The lemon fruit is used mainly as a fresh fruit, but it is also processed to make juices, jams, jellies, molasses, etc. [41]. Fresh lemon fruit can be kept for several months, maintaining their levels of juice, vitamins, minerals, fibre, and carbohydrates. The vitamin C (ascorbic acid) content in lemon fruits and juices decreases during storage and industrial processing. The factors lowering this content are: oxygen, heat, light, time, storage temperature and storage duration. To prevent the reduction in the ascorbic acid levels and antioxidant capacity of both the lemon fruit and lemon juice, they should be kept at 0–5 °C and protected from water loss by proper packaging, with high relative humidity during distribution. Under such conditions, lemon products show a good retention of vitamin C and antioxidant capacity [41,74].

*C. limon* peel is rich in pectin, which is used in a wide range of food industrial processes as a gelling agent, including the production of jams and jellies, and as thickener, texturizer, emulsifier and stabilizer in dairy products. Due to its jellifying properties, the pectin is also used in pharmaceutical, dental and cosmetic formulations [75].

Lemon juice is used as an ingredient in beverages, particularly lemonade and soft drinks, and in other foods, such as salad dressings, sauces, and baked products. Lemon juice is a natural flavouring and preservative, and it is also used to add an acidic, or sour, taste to foods and soft drinks [41,76].

*C. limon* is the most suitable, being free from pesticide residues, raw material for enhancing the flavour of liqueurs, e.g., “limoncello”, the traditional liqueur of Sicily. It is made by the maceration of lemon peel in ethanol, water and sugar [41,76].

Currently, the essential oil from lemon, i.e., pure isolated linalol and citral, are used mainly as a flavouring and natural preservative due to their functional properties (antimicrobial, antifungal, etc.) [52,53]. In particular, they are often used to extend the short shelf-life of seafood products and in the production of some types of cheese because they significantly reduces populations of microorganisms, especially those from the family *Enterobacteriaceae* [41,76].

## 9. Cosmetological Applications

*C. limon* fruit extracts and essential oil, as well as the active compounds isolated from these raw materials, have become the object of numerous scientific studies aimed at proving the possibility of their use in cosmetology. Lemon-derived products have long been credited with having a positive effect on acne-prone skin that is easily affected by sunburn or mycosis. In this regard, traditional uses of this raw materials are known in various parts of the world. In Tanzania, the fruit juice of *C. limon* is mixed with egg albumin, honey and cucumber, and applied to the skin every day at night to smooth the facial skin and treat acne [77]. Juice from freshly squeezed fruit of *C. limon* mixed with olive oil is used as a natural remedy for the treatment of hair and scalp disorders in the West Bank in Palestine [78]. Currently, knowledge of the cosmetic activity of *C. limon* is constantly expanding.

*C. limon* essential oil shows antibiotic and flavouring properties, and for this reason it is used in formulations of shampoos, toothpaste, disinfectants, topical ointments and other cosmetics [41].

Scientific studies have shown a significant antioxidant effect of *C. limon* fruit extracts, which is the reason they are recommended for use in anti-ageing cosmetics [8,48]. The use of different carriers for *C. limon* extracts (e.g., hyalurosomes, glycerosomes) in cosmetics production technology contributes to an even greater inhibition of oxidative stress in skin-building structures, including keratinocytes and fibroblasts (Table 9) [79]. In addition, vitamin C from *C. limon* is used as an ingredient in specialized dermocosmetics. Its external use increases collagen production, which makes the skin smoother and more tense. It is used in anti-aging products, to reduce shallow wrinkles, and as a synergistic antioxidant in combination with vitamin E [48].

The ingredients of *C. limon* essential oil (including citral, β-pinene, D-limonene), due to the inhibiting activity of tyrosinase and the inhibition of L-dihydroxyphenylalanine (L-DOPA) oxidation, have a depigmenting effect [80]. In addition, the essential oil has been proven to support the penetration of lipids as well as water-soluble vitamins. It can be used as a promoter of penetration of active substances through the skin [81]. Moreover, besides the direct effect on the skin, the essential oil can also be used as a natural preservative and as a corrigent in cosmetic products. Studies have confirmed its antibacterial and fungistatic effects (Table 9) [7,52,53].

Furthermore, *C. limon* pericarp extracts, too, exhibit scientifically proven activity that helps to accelerate the regeneration of diabetic wounds. In addition, the essential oil derived from *C. limon* pericarp has shown anti-inflammatory, anti-allergic and slimming properties [49,59,60,62].

According to the CosIng Database (Cosmetic Ingredient Database), *C. limon* can be used in twenty-three forms. It can be used in cosmetics in the form of oils obtained from various organs, in the form of extracts, hydrolates, powdered parts of the plant, wax and juice [27]. The most common activity defined by CosIng for the raw material of this species is to keep the skin in good condition, to improve the odour of cosmetic products, and to mask the smell of other ingredients of cosmetic preparations [27]. The approved forms of raw materials and their potential effects, as well as their use as corrigents, presented in the CosIng Database, are summarized in Table 10 [27].

*C. limon* essential oil has been used since the 18th century in the production of the famous ‘Eau de Cologne’. In aromatherapy, it is used to treat numerous diseases and lifestyle-related ailments: hypertension, neurosis, anxiety, varicose veins, arthritis, rheumatism and mental heaviness. It also alleviates symptoms characteristic of menopause. *C. limon* essential oil is also used in aromatherapy massages to relax muscles, and for calming down and deep relaxation [21].

*C. limon* fruit extracts and essential oil should not be used in high concentrations in baths or directly on the skin. Recent studies have shown that D-limonene contained in the oil has an allergenic and irritating effect on the skin. It may cause cross-allergy with balsam of Peru. After applying cosmetics containing *C. limon* oil, it is forbidden to expose the skin to sunlight*. C. limon* essential oil contains photosensitizing compounds belonging to the linear furanocoumarin group. The lemon pericarp contains: bergapten, phellopterin, 5- and 8-geranoxypsoralen, and the essential oil contains: bergapten, imperatorin, isopimpinellin, xanthotoxin, oxypeucedanin and psoralen [21,82].

The International Fragrance Association (IFRA) has issued restrictions on the use of *C. limon* essential oil. In preparations remaining on the skin, the concentration of that oil should not exceed 2%. In addition, *C. limon* essential oil should not be used in preparations remaining on skin exposed to UV rays. They should not contain more than 15 ppm of bergapten [83].

## 10. Plant Biotechnological Studies on *C. limon*

Plant biotechnology creates opportunities for the potential use of plant in vitro cultures in the pharmaceutical, cosmetics and food industries. In vitro cultures can be a good alternative to plants growing in vivo. Plant biotechnology enables control and optimization of the conditions for conducting in vitro cultures to increase the accumulation of active compounds. It facilitates, among other things, optimization of the culture medium, including the concentration of plant growth and development regulators, the use of elicitors (stressors), the selection of highly productive cell lines and genetic transformations. In vitro cultures can also be used in plant propagation (micro-propagation process) [84].

*C. limon* cultures in vitro have thus far been the subject of research concerned with the development of micropropagation protocols. They have focused on the selection of plant growth regulators (PGRs) that induced shoot and root production in in vitro cultures. In 2012, biotechnological research on the micropropagation of *C. limon* was performed by Goswami et al. [85] from SKN Rajasthan Agricultural University in India. Shoot cultures were propagated from plant nodes on a Murashige and Skoog (MS) medium [86] containing different types and concentrations of PGRs. The maximum number of shoots and shoot regenerations was observed at a low level of 6-benzyladenine (BA) −0.1 mg/L, or kinetin −0.5 mg/L. Shoot proliferiation was also observed in combinations of PGRs such as BA and 1-naphthaleneacetic acid in concentrations of 0.1 mg/L each. With an increase in BA concentration in MS medium, shoot proliferation decreased. Regenerated shoots showed root induction on MS basal medium or on MS medium containing 1.0 mg/L of indole-3-butyric acid.

Another biotechnological study on *C. limon* was carried out in the Department of Citriculture in Murcia (Spain) [87]. The researchers studied organogenesis and made histological characterization of mature nodal explants of two important cultivars of *C. limon*—‘Verna 51’ and ‘Fino 49’. The highest number of buds per regenerating explant was obtained on the MS medium in comparison with the Woody plant medium [88]. The presence of 1–3 mg/L BA, in combination with 1 mg/L of 1-gibberellic acid (GA) in the culture medium, was essential for the development of adventitious buds. The lowest extent of organogenesis was observed when BA was used in the medium without GA [87].

## 11. Conclusions

The presented review proves that *C. limon* is a very attractive object of different scientific studies. The *C. limon* fruit is a raw material that can be used in different forms, e.g., extracts, juice and essential oil. The rich chemical composition of this species determines a wide range of its biological activity and its being recommended for use in phytopharmacology. The studies have focused on the essential oil and its main active compound—D-limonene. Extracts from *C. limon* fruits are rich in flavonoids such as naringenin and hesperetin.

Current pharmacological studies have confirmed the health-promoting activities of *C. limon*, especially its anti-cancer and antioxidant properties. *C. limon* also finds increasing application in cosmetology and food production.

There has been some biotechnological research aimed at developing effective in vitro micropropagation protocols for *C. limon*.

## Figures and Tables

**Figure 1 plants-09-00119-f001:**
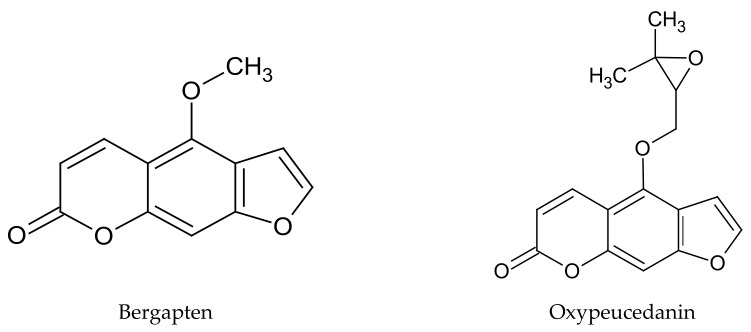
Chemical structure of selected linear furanocoumarins, determining the photosensitizing effect of *C. limon.*

**Figure 2 plants-09-00119-f002:**
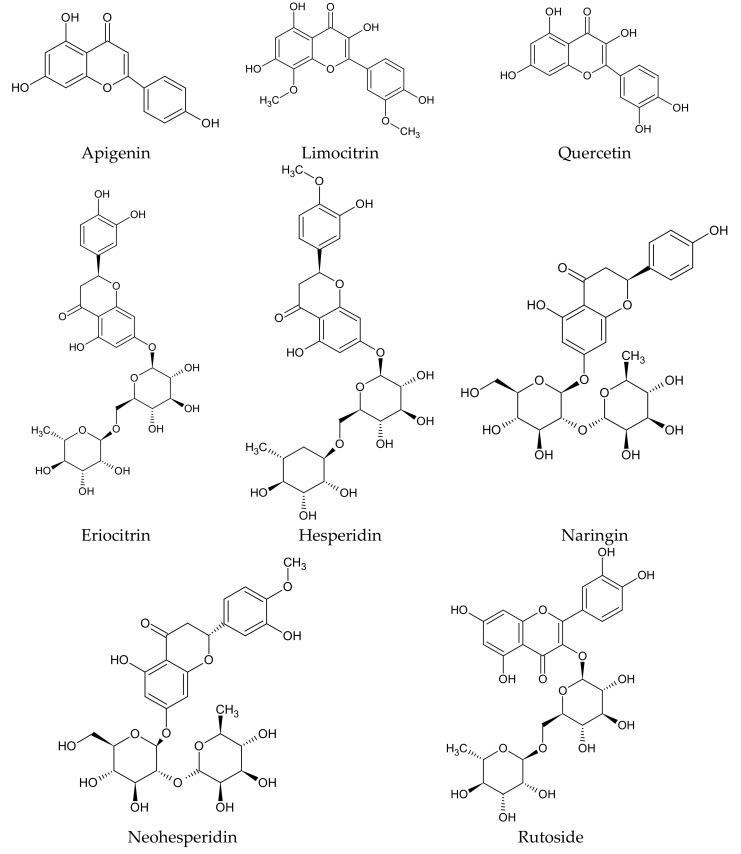
Chemical structure of flavonoids characteristic of *C. limon.*

**Figure 3 plants-09-00119-f003:**
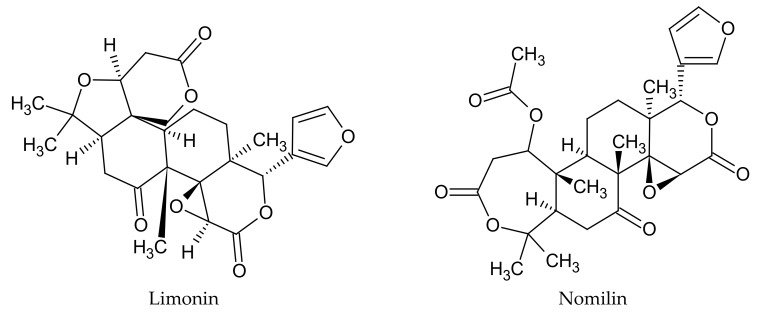
Chemical structure of limonoids characteristic of *C. limon.*

**Figure 4 plants-09-00119-f004:**
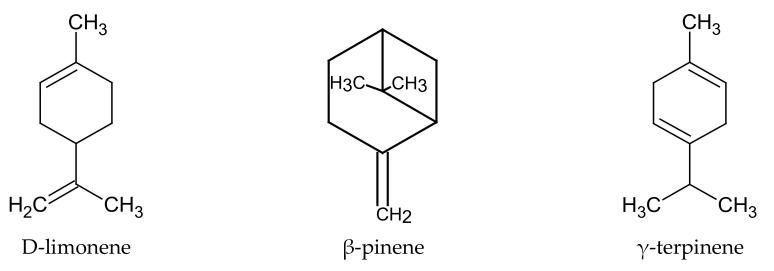
Chemical structure of selected terpenoids characteristic of *C. limon* essential oil.

**Table 1 plants-09-00119-t001:** *C. limon* cultivars.

Cultivar Name	Origin	Cultivation	Characteristic
*C. limon* ‘Bearss’ (*C. limon* ‘Sicilian’, Bearss lemon)	Florida	Florida, Brazil	It grows quickly and is very productive. It has aromatic flowers, juicy fruit and a high sensitivity to low temperatures.
*C. limon* ‘Berna’—(*C. limon* ‘Verna’, Verma lemon)	Spain	Spain	The specimens are large, without spines. It bloom two to three times a year. Fruits from individual harvests differ in properties and are called ‘Cosecha’ (main collections), ‘Secundus’ and ‘Rodrejos’.
*C. limon* ‘Eureka’ (Eureka lemon)	California, Sicily	Mediterranean Basin, California, Australia, Argentina, South Africa, Israel	Oblong fruit with a smooth skin and a small amount of stones. Flowers of a pink shade.
*C. limon* ‘Femminello’	Italy	Italy	A very productive variety. It blooms and bears fruit throughout the year.
*C. limon* ‘Genova’ (*C. limon* ‘Genoa’)	Italy	California, Florida, Chile	Spike-free trees, resistant to cold with dense foliage. Yellow fruits with a marked tip have a smooth and thin pericarp.
*C. limon* ‘Interdonato’	Italy	Italy	It has large, oblong, cylindrical pointed fruit. Pericarp strongly adheres to the fruit; it is thin, smooth, and yellow. With few seeds.
*C. limon* ‘Lisbon’	Portugal	California, Arizona, Australia, Uruguay, Argentina	It has long spines, thick skin, pink flowers, and pale-yellow flesh.
*C. limon* ‘Monachello’	Italy	Italy	The main advantage of this variety is high resistance to the disease caused by *Phoma tracheiphila*.
*C. limon* ‘Primofiori’ (*C. limon* ‘Fino’, *C. limon* ‘Mesero’, *C. limon* ‘Blanco’	Spain	Spain	A productive variety with spines. Fruits have a spherical or oval shape, with a small wart at the end.
*C. limon* ‘Santa Teresa’ (*C. limon* ‘Feminello Santa Teresa’, *C. limon* ‘Italian’)	Italy	Italy, North-West Argentina, Turkey	Pericarp, contains a large amount of essential oil. The fruit contains a large amount of juice. This variety is resistant to storage and transport.
*C. limon* var. Variegata (*C. limon* ‘Eureka’ var. Variegated, Pink-fleshed lemon	California	California	Established as a result of the intrinsic mutation of *C. limon* ‘Eureka’ in 1931. It has pulp and juice of a pink shade. The fruits are yellow with green stripes and variegated leaves.
*C. limon* ‘Villafranca’	Sicily	Florida, Israel, North-West Argentina	It has pulp and juice of a pink shade. The fruit is yellow with green stripes.

**Table 2 plants-09-00119-t002:** Hybrids of *C. limon*.

Name	Origin	Characteristic
*C. limon* ‘Lemonime’	hybrid *C. limon* and *C. aurantifolia*	It has fruit larger than limes (*C. aurantifolia*).
*C. limon* ‘Lumia’	hybrid *C. maxima* and *C. medica*, subsequently hybridized with *C. limon*	The fruit resembles a pear. It can reach large sizes.
*C. limon* ‘Ponderosa’	hybrid *C. limon* and *C. medica*	Fruits with a pear-shaped and thick pericarp.
*C. limon* ‘Volkamer’	hybrid *C. limon* and *C.aurantium*	Specimens smaller than *C. limon*. The fruit has few seeds and a thick, rough, light reddish pericarp. The flesh and juice are yellow-red. The hybrid is resistant to many diseases.

**Table 3 plants-09-00119-t003:** Composition of *C. limon* fruits extracts.

Group of Compounds	Part of Fruit	Metabolites
Flavonoids	Whole fruit (pulp, seed and peel)	flavonones: eriocitrin, eriodiktyol, hesperidin, naringin, neoeriocitrin, neohesperidin
		flavones: apigenin, diosmetin, diosmin, homoorientin, luteolin, orientin, vitexin
		flavonols: isoramnethin, quercetin, limocitrin, rutoside, spinacetin
Limonoids	Whole fruit (pulp, seed and peel)	limonin, nomilin
Phenolic acids	Whole fruit (pulp, seed and peel)	dihydroferulic acid, p-hydroxybenzoic acid, 3-(2-hydroxy-4-methoxyphenyl)propanoic acid, synapic acid
Carboxylic acids	Whole fruit (pulp, seed and peel)	citric acid, galacturonic acid, glucuronic acid, glutaric acid, homocitric acid, 3-hydroxymethylglutaric acid, isocitric acid, malic acid, quinic acid
Coumarins	Whole fruit (pulp, seed and peel)	citropten (5,7-dimethoxycoumarin), scopoletin
Furanocoumarins	Whole fruit (pulp, seed and peel)	bergamottin
Amino acids	Whole fruit (pulp, seed and peel)	L-alanine, L-arginine, L-asparagine, L-aspartic acid, dimethylglycine, glutamic acid, L-phenylalanine, DL-proline, L-tryptophan, L-tyrosine, L-valine
Carbohydrates	Peel	monosaccharides: arabinose, fructose, β-fructofuranose, β-fructopyranose, galactose, glucose, mannose, myoinositol, rhamnose, scylloinositol, xylose
	Whole fruit (pulp, seed and peel)	disaccharides: sucrose
Vitamins and theirsmetabolites	Whole fruit (pulp, seed and peel)	choline, pantothenic acid, trigoneline, vitamin C
Macroelements	Pulp and peel	calcium (Ca), magnesium (Mg), phosphorus (P), potassium (K), sodium (Na)

**Table 4 plants-09-00119-t004:** Composition of *C. limon* juice.

Group of Compounds	Metabolites
Flavonoids	flavonones: hesperidin, naringin flavones: apigenin, chrysoeriol, diosmetin, luteolin flavonols: isoramnethin, quercetin, rutoside dihydroxyflavonols: dihydroxyisoramnethin-7-O-rutinoside
Phenolic acids	ferulic acid, synapic acid
Vitamins	vitamins: C (53 mg/L), A, B_1_, B_2_, B_3_

**Table 5 plants-09-00119-t005:** Composition of oil from *C. limon* seeds.

Group of Compounds	Metabolites
Fatty acids	arachidonic acid, behenic acid, lignoceric acid, linoleic acid, linolenic acid, oleic acid, oleopalmitic acid, palmitic acid, stearic acid
Tocopherols	α-tocopherol, β-tocopherol, γ-tocopherol, δ-tocopherol
Carotenoids	β-carotene, β-cryptoxanthin, lutein

**Table 6 plants-09-00119-t006:** The chemical composition of the essential oil of the *C. limon* pericarp and leaf.

Group of Compounds	Essential Oil	Metabolites
Terpenoids	essential oil of the *C. limon* pericarp	limonene (69.9%), p-mentha-3,8-diene (18.0%), β-pinene (11.2%), γ-terpinene (8.21%), myrcene (4.4%), sabinene (3.9%), myrcene (3.1%) geranial (E-citral, 2.9%), neral (Z-citral, 1.5%), linalool (1.41%), α-pinene (1.1%), α-thujene (1.1%), β-bisabolene (0.5%) (E)-β-ocimene (0.4%), citronellol (0.3%), geraniol (0.2%), β-caryophyllene (0.2%), trans-muurala-4(14),5-diene (0.2%), α-terpinene (0.1%), terpinolene (0.1%), nonanal (0.1%), eucalyptol (0.1%); other terpenes below 0.06%: α-bisabolol, camphene, citronellal, citronellyl acetate, β-curcumene, γ-curcumene, p-cymene, 7-epi-sesquithujene, α-farnesene, α-felandren, cis-limonene, trans-limonene, octanal, octanal acetate, terpinen-4-ol, β-santalene, zonarene
	essential oil of the *C. limon* leaf	limonene (31.5%), sabinene (15.9%), citronellal (11.6%), linalool (4.6%), neral (4.5%), geranial (4.5%), (E)-β-ocimene (3.9%), myrcene (2.9%), citronellol (2.3%), β-caryophyllene (1.7%), terpne-4-ol (1.4%), geraniol (1.3%), α-pinene (1.2%),γ-terpinene (0.9%), sylvestrene (0.6%), α-terpineol (0.6%), isogeranial (0.4%), β-bisabolene (0.3%), germacrene B (0.3%), isospathulenol (0.3%), α-terpinene (0.2%), terpinolene (0.2%), isopulegol (0.2%), γ-terpineol (0.2%), decanal (0.2%), δ-elemene (0.2%), α-humulene (0.2%), α-cadinol (0.2%), *epi*-α-bisabolol (0.2%) *cis*-p-menth-2-en-1-ol (0.1%), isoneral (0.1%), γ-muurolene (0.1%), spathulenol (0.1%)
Furano-coumarins	essential oil of the *C. limon* pericarp	aprindine, bergamottin, bergapten, byakangelicol, byakangelicin, epoxybergamottin, 5- and 8-geranoxypsoralen, 8-geranyloxypsoralen, heraclenin, imperatorin, isoimperatorin, isopimpinellin, xanthotoxin, oxypucedanin, phellopterin, psoralen
Coumarins	essential oil of the *C. limon* pericarp	citropten, 5-geranyloxy-7-methoxycoumarin, herniarin, 5-isopentenyloxy-7-methoxycoumarin

**Table 7 plants-09-00119-t007:** Biological activity of *C. limon* fruit extracts confirmed by scientific research.

Activity	Mechanism of Action	References
Anticancer activity	- Inhibition of the proliferation of cancer cells; - Activation of “TRAIL”-apoptopic cell death; - Inhibition of tumour growth in chronic yelogenous leukaemia (CML); - Antioxidant action and induction of apoptosis in MCF-7 cells (breast adenocarcinoma cells) (*C. limon* seed extract).	[9,47]
Antioxidant activity	- Augmention of antioxidant cellular defences via ERK/Nrf2 signalling pathway.	[8,48]
Anti-inflammatory activity	- Inhibition of NF-κB factor, nitric oxide (II) synthase (iNOS), induced cyclooxygenase (COX-2) (hesperidin, hesperitin); - Down-regulation of TLR-signaling pathway (limonin).	[8,49,50,52]
Antibacterial activity	- Inhibiting activity against Gram-positive bacteria: *Enterococcus faecalis*, *Bacillus subtilis;* - Inhibiting activity against Gram-negative bacteria: *Salmonella typhimurium*, *Shigella sonnei*, *Helicobacter pylori.*	[5,7]
Antifungal activity	- Inhibiting activity against *Candida glabrata* strains.	[7]
Antiviral activity	- Inhibition of replication of *Herpes simplex.*	[57]
Anti-allergic activity	- Inhibition of histamine secretion in peritoneal cells of rats.	[1,59]
Hepatoregenerative activity	- Normalization of alanine aminotransferase (ALAT), alkaline phosphatase (ALP) and bilirubin; - Reduction in malonic dialdehyde (MDA), lipid peroxidation, superoxide dismutase (SOD) and catalase.	[10]
Prevention of diabetes and treatment of its symptoms	- Inhibition of gluconeogenesis (naringenin, hesperitin); - Inhibition of gluconeogenesis (naringenin, hesperitin); - Reducing wound-healing time; - Increasing tissue growth rate, collagen synthesis, and protein and hydroxyproline concentration.	[1,60]
Anti-obesity activity	- Lowering blood lipids; - Reducing the levels of insulin, leptin and adiponectin in the blood.	[1,62]
Effects on the cardiovascular system	- Limiting myocardial damage (naringenin); - Decreasing blood fibrinogen; - Lowering blood pressure in people with hypertension.	[1,11,63]
Effects on the nervous system	- Strengthening short-term memory.	[67]
Effects on the respiratory system	- Treatment of chronic pneumonia (naringenin).	[68]
Effects on the skeletal system	- Increasing bone density; - Decreasing osteoclast activity; - Decreasing TRAP-positive multinucleated cell numbers (nomilin); - Decreasing bone resorption activity (nomilin); - Down regulation osteoclast-specific genes (NFATc1 and TRAP mRNA levels) (nomilin).	[66,69]
Treatment of menstrual disorders	- Period induction in cases of irregular menstrual cycles.	[5]

**Table 8 plants-09-00119-t008:** Biological activity of *C. limon* essential oil confirmed by scientific research.

Activity	Mechanism of Action	References
Anticancer activity	- Stimulation of apoptosis of colorectal cancer cells.	[70]
Anti-inflammatory activity	- Inhibiting cell migration; - Inhibition of cytokine production; - Inhibition of inflammation mediator (D-limonene); - Inhibition of leukocyte chemotaxis (D-limonene); - Interaction with 5-lipoxygenase, TNF-α (tumour necrosis factor), IL-6 (interleukin-6).	[49,52]
Antibacterial activity	- Inhibiting activity against Gram-positive bacteria: *Staphylococcus capitis, Micrococcus luteus, Bacillus subtilis*; - Inhibiting activity against Gram-negative bacteria: *Pseudomonas fluorescens, Escherichia coli*.	[21,52,53]
Antifungal activity	- Inhibiting activity against: *Aspergillus niger*, *Saccharomyces cerevisiae*, *Candida parapsilosis* strains (D-limonene, β-pinene, citral).	[21,52,56]
Antiviral activity	- Inhibition of the virus *Herpes simplex*.	[57]
Antiparasitic activity	- Treatment of schistosomiasis caused by *Schistosoma mansoni* (D-limonene); - Inhibitory effect on *Sarcoptes scabiei* development.	[58]
Anticaries activity	- Inhibiting growth of *Streptococcus mutans* and its adhesion to enamel; - Inhibition of glucosyltransferase transcription and enzymatic activity.	[54]
Hepatoprotective and detoxification activity	- Stimulation of liver detoxification by activation of cytochrome P_450_ and liver enzymes (glutathione S-transferase) in chronic liver poisoning.	[71]
Diabetes prevention	- Decreased glycolized haemoglobin (D-limonene); - Decreased gluconeogenesis enzymes: glucose-6-phosphatase and fructose-1,6-biphosphatase (D-limonene); - Decreased blood glucose (D-limonene).	[61]
Anti-obesity activity	- Lowering cholesterol and preventing fat deposits (D-limonene); - Equalization of blood sugar (D-limonene); - Regeneration of pathological changes in the liver and pancreas.	[72]
Effect on the digestive system	- Increased gastric motility and reduction of nausea (D-limonene); - Neutralization of stomach acids (D-limonene); - Relief of gastric reflux (D-limonene); - Increased bile flow.	[21]
Lipolytic and cholesterol-lowering activity	- Reducing the level of triglycerides, LDL and increasing the level of HDL cholesterol in the blood; - Lowering cholesterol and arachidonic acid levels by stimulating liver enzymes and cytochromes; - Lipolytic effect (γ-terpinene and p-cymene).	[1,21,72]
Effects on the nervous system	- Inhibitory effect on pain receptors similar to that of indomethacin and hyoscine (D-limonene); - Sedative and anxiolytic effect by activating serotonin and dopamine receptors.	[73]

**Table 9 plants-09-00119-t009:** Biological activity of *C. limon* fruit extracts, essential oil and its ingredients compounds significant from the cosmetics point of view, confirmed by scientific research.

Activity	Extracts and Compounds Tested	Mechanism of Action	References
Antioxidant activity	*C. limon* essential oil	Strong lipid peroxidation reduction and free radical reduction effect in vitro and in vivo.	[79,80]
	*C. limon* var. *pompia* fruit extracts	Extract enclosed in hyalurosomes and glycerosomes reduces oxidative stress caused by hydrogen peroxide and the viability of keratinocytes and fibroblasts.	
Depigmenting activity	Essential oil ingredients (e.g., citral, β-pinene, D-limonene)	Essential oil components show tyrosinase inhibitory activity. Mixture of essential oil ingredients has a stronger inhibitory effect due to their synergistic effect.	[80]
Effect of increasing the penetration of substances	*C. limon* essential oil	Acc. to in vitro study on human epidermal cells (SkinEthic), *C. limon* essential oil increased the penetration of α-tocopherol. Modification of TEWL (Trans Epidermal Water Loss) was transient. *C. limon* essential oil enhanced the penetration of locally administered lipids and water-soluble vitamins.	[81]
Preservative effect in cosmetics	*C. limon* essential oil	Antibacterial activity and increasing the fungistatic effect of synthetic preservatives.	[7,52,53]

**Table 10 plants-09-00119-t010:** *C. limon* in cosmetic products according to CosIng.

The Form	Activity
*C. limon* (lemon)/*Fucus serratus* extract	skin conditioning
*C. limon* bud extract	humectant, skin conditioning
*C. limon* flower water	humectant, skin conditioning
*C. limon* flower/leaf/stem extract	masking, skin conditioning, tonic
*C. limon* flower/leaf/stem oil	masking
*C. limon* fruit extract	masking, skin conditioning
*C. limon* fruit oil	astringent, tonic
*C. limon* fruit powder	skin conditioning
*C. limon* fruit water	masking, skin conditioning
*C. limon* juice	skin conditioning, tonic
*C. limon* juice extract	tonic
*C. limon* juice powder	skin conditioning, tonic
*C. limon* leaf extract	perfuming
*C. limon* leaf oil	perfuming, masking
*C. limon* leaf/peel/stem oil	skin conditioning
*C. limon* peel	masking, skin conditioning
*C. limon* peel cera/*C. limon* peel wax	skin conditioning
*C. limon* peel extract	emollient, skin conditioning, skin protecting, tonic
*C. limon* peel oil	masking, perfuming, skin conditioning
*C. limon* peel powder	absorbent, viscosity controlling
*C. limon* peel water	skin conditioning
*C. limon* seed oil	masking, perfuming, skin conditioning

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
