# Peer review of "Citrus limon (Lemon) Phenomenon—A Review of the Chemistry, Pharmacological Properties, Applications in the Modern Pharmaceutical, Food, and Cosmetics Industries, and Biotechnological Studies"

_plants, 2020, doi:10.3390/plants9010119_

Round 1

Reviewer 1 Report

The paper is a review dealing with important botanical, chemical and pharmacological characteristics of Citrus limon (lemon). The manuscript is well written and offers a comprehensive report of the genus Citrus followed by information on the chemical composition and biological activities of the main raw materials obtained from C. limon.  A careful description of the preparations used in the traditional medicine as well as cosmetic and culinary properties is reported. An interesting list of scientifically proven therapeutic activities of C. limon  including anti-inflammatory, antimicrobial, anticancer and antiparasitic is also included. The paper deserves publication in Plants since it is certainly of interest for both the general readers and the researchers in the field. Since the metabolomics approach has been recently used (as for many other natural substrates) also for lemon (and derivatives) study a brief referenced description of these studies could even improve the final outcome of this already interesting paper.

Reviewer 2 Report

In this review article, the authors systematize valuable scientific work on the study of Citrus limon, its chemical composition and pharmaceutical properties, and their applications to the pharmaceutical and chemical, food and cosmetic industries. Finally, the review emphasizes articles correlated with biotechnological studies.
I find the review article is updating the existing literature and providing valuable assistance to researchers.

Reviewer 3 Report

The manuscript mainly describes the genus, botanical characteristics, chemical composition, and biological activity of Citrus limon. Among them, description about chemical composition and biological activity were not enough. I listed the points that authors should add more description. This manuscript needs major revision for acceptance.

1) Authors should provide absolute stereochemistry of all compounds. In addition, described chemical structures were not well-ordered. Bonds and atoms must not be overlapped.

2) Described compounds from C.limon were very limited. Authors should add more compounds including minor constituents and recently reported compounds.

3) For Table 3 or Figure, add new columns or border line to identify each compound were isolated from which parts (seeds, peels, or something.)

4) For section ‘Biological activity of C. limon raw materials’ , are there no data about bioactive constituents? I think, there are many paper mentions about bioactive constituents.

5) The bioactivities without chemical structure were not interesting because it is difficult to explain it’s advantage. For example, flavonoids and limonoids were reported as the constituents of other plants. Please add a points of uniqueness of C.limon.

Round 2

Reviewer 3 Report

The revised manuscript is improved at the points of chemical composition and biological activity. I think, this manuscript is acceptable very minor changes.

 In Figure 3 and 4, atoms and bonds should not overlap.

 In Table 10, Authors need not wright 'Citrus limon'. It should be  'C.limon'